# The Genome-Wide Profiling of Alternative Splicing in Willow under Salt Stress

Xue Wang [1,†], Longfeng Gong [1,†], Junkang Zhang [2], Lei Wang [2], Di Wu [2] and Jichen Xu [1,2,*]

1 State Key Laboratory of Tree Genetics and Breeding, College of Biological Sciences and Technology, Beijing Forestry University, Beijing 100083, China
2 National Engineering Research Center of Tree Breeding and Ecological Restoration, College of Biological Sciences and Technology, Beijing Forestry University, Beijing 100083, China
* Correspondence: xujichen@bjfu.edu.cn; Tel.: +86-10-6233-6628
† These authors contributed equally to this work.

**Abstract:** Alternative splicing (AS) is an important post-transcriptional regulatory model that can change the normal transcript expression level and possibly result in protein diversity. In this study, we conducted the full-length transcript sequencing of *Salix matsudana* Koidz 9901 leaves under salt treatment using the PromethION platform. A total of 4786 AS genes (9307 AS events) were determined, accounting for 7.45% of all the transcribed genes. Of them, intron retention (IR) events accounted for the most AS events (46.05%), followed by alternative 3′ splice sites (A3SS). During salt stress, the percentage of IR events decreased, and the percentage of the others increased. Statistical results showed that 5′GG was the most common motif at the 5′ end of the intron in the AS events, and GG3′ was the most common motif at the 3′ end. 5′GG-AG3′ was the most common splice mode in the AS events. The occurrence of AS events was significantly related to the exon number, exon length, intron length, GC content, and expression abundance of the genes. During salt stress, the number of AS genes gradually increased, and they mainly participated in purine and chlorophyll metabolism, RNA transport, and autophagy. Meanwhile, the AS sites of the gene increased during salt treatment, indicating the complexity of the AS events by salt stress. A comparison of differentially expressed genes (DEGs) and differentially alternative splicing (DAS) genes during salt stress revealed that they had a different mechanism of gene expression regulation when subjected to salt stress. These results expand our knowledge of AS events and shed light on and improve our understanding of plant resistance to salt tolerance in willow.

**Keywords:** *Salix matsudana*; alternative splicing; salt stress; transcriptome; intron model

## 1. Introduction

In higher plant genomes, most genes have multiple introns, which are excised via the spliceosome during mRNA maturation. The intron pattern is mostly conserved, with the motifs flanking it being generally 5′GT-AG3′ [1]. In some cases, the transcripts had different splicing sites and resulted in different mRNA products called alternative splicing (AS). More studies revealed that AS is an important gene expression regulation mode and is widely involved in plant growth and adversity resistance [2–4]. The RNA sequencing of 28 soybean samples at different developmental stages showed that more than 63% of multiexon genes underwent AS [5]. The transcriptome analysis of moso bamboo shoot samples from different growth periods showed that 60.74% of the genes underwent AS, and the winter bamboo shoot sample had the most splicing events [6]. An investigation of wood formation in *Populus* and *Eucalyptus* showed that 28.3% and 20.7% of the highly expressed transcripts in the xylem underwent AS events [7]. Notably, 42% of the AS events resulted in frameshift mutations, and 25% and 26.8% of the AS events in *Populus* and *Eucalyptus* resulted in protein domain variation, respectively. The proportion of AS events for genes homologous in both tree species was 28%, and 71 conserved AS events were

further confirmed. An analysis of *cassava* genes under cold and drought stress revealed 38,164 AS events, and among those, 3292 and 1025 events were regulated by cold and drought stress, respectively [8]. A premature termination codon was introduced in 58.5% of the AS transcripts.

AS events seriously affect the expression level of normal transcripts and are an important post-transcriptional regulation mode. Some splicing variants are endowed with new functions and are involved in plant growth and stress resistance [9]. The flowering locus T2 (*FT2*) gene of *Brachypodium distachyon* has two age-dependent splicing variants, FT2α and FT2β; the FT2β protein lacks the PEBP domain and cannot interact with 14-3-3 and FD proteins to assemble the flowering initiation complex. The ectopic expression of FT2α in *Brachypodium distachyon* resulted in stagnant nutrient growth and early flowering, whereas the transgenic lines with FT2β overexpression had delayed flowering and tasseling [10]. The *OsHsfA2a* gene encodes a rice heat shock transcription factor, which can produce six isoforms (*OsHsfA2a-1~OsHsfA2a-6*) through AS. The six isoforms have tissue-specific expression patterns in rice and show different stress response patterns. For example, *OsHsfA2a-2* expression was rapidly upregulated under heat stress, salt stress, and ABA treatment. *OsHsfA2a-6* showed the highest transcript level under drought stress, while *OsHsfA2a-3* expression was gradually reduced [11]. Under drought and salt stress, *OsDREB2B* produces two transcripts. *OsDREB2B1* is a nonfunctional isoform containing a premature termination codon (PTC) and is predominant under normal conditions. *OsDREB2B2*, which is significantly induced under drought and salt stresses, encodes a transcriptional activation protein that significantly enhances plant tolerance against stress [12]. A study analyzed two tomato genotypes against drought and low nitrate stress independently or in combination. Several genes in the TOR pathway, phytohormone metabolism, transporter and signaling, abscisic acid, ethylene, and growth hormone-related pathways suffered from differential AS processes under stress conditions. Further functional characterization of these AS events is expected to facilitate the development of superior tomato varieties tolerant to environmental stress [13].

AS events in plants are categorized into four types, namely exon skipping (ES), intron retention (IR), alternative 5′ splicing site (A5SS), and alternative 3′ splicing site (A3SS) [14]. A survey on *Populus tomentosa Carr* under regular conditions showed that 44.72% of the AS events were of the IR type, and 26.14%, 19.60%, and 9.54% of the AS events were of the A5SS, A3SS, and ES types, respectively. Even the ratio of the four AS types varied during heat treatment [15].

*Salix matsudana* Koidz 9901 is a highly salt-tolerant willow variety that is widely planted in saline–alkali land. The study here aimed to evaluate the AS events and patterns in willow genes under salt treatment to reveal the characteristics of willow AS genes and their effects on salt resistance. This study provides new insights into plant stress resistance and presents more evidence for further breeding programs.

## 2. Materials and Methods

### 2.1. Plant Materials and Treatment

The cuttings of *S. matsudana* Koidz 9901 were planted in pots (30 cm in height × 20 cm in diameter) with a mixture of nutrient soil (Chenchen Biotechnology Co., Shanghai, China) and vermiculite (1:1), and cultured in a light incubator (25 °C, 120 mol·m$^{-2}$·s$^{-1}$, 16 h/8 h light/dark period). After growth for 3 months, they were treated with 300 mM of NaCl solution once until the sample was harvested on the 6th day (500 mL per pot was well saturated in the medium, and then the leaked solution was discarded).

### 2.2. Physiological Index Test

Mature willow leaves were collected after salt treatment for 0 d, 3 d, and 6 d for the physiological index tests. Three biological replicates were conducted for each treatment. All data were analyzed using ANOVA and Student's *t*-test for the significant difference evaluation with SPSS 24.0 software.

Relative electrical leakage (REL): Fresh leaves (0.1 g) were cut into small pieces (0.5 cm × 0.5 cm), soaked in 30 mL of deionized water, and shaken at 25 °C at 180 rpm overnight. The conductivity was measured as R1 with an electrical conductivity meter (DDS-11C). The samples were then autoclaved at 121 °C for 20 min and shaken overnight again. The conductivity was then measured as R2. The leaf REL was calculated as R1/R2 [16].

Malondialdehyde content (MDA): Fresh leaves (0.1 g) were ground in 1 mL of 10% trichloroacetic acid. The homogenized mixture was centrifuged at 12,000 rpm for 5 min. The supernatant was collected and mixed with 1 mL 0.6% thiobarbituric acid, boiled for 15 min, and then centrifuged at 12,000 rpm for 5 min. The light absorbance of the supernatant was measured at 450 nm, 532 nm, and 600 nm wavelengths. The MDA content was calculated as [6.45 × (OD532 − OD600) − 0.56 × OD450] × total extract volume/fresh weight of the sample [16].

Chlorophyll content: Fresh leaves (0.1 g) were placed in 10 mL of dimethyl sulfoxide (DMSO) at 25 °C in the dark for 2 days. The light absorption value of the extract solution was measured at wavelengths of 663 nm and 645 nm. The chlorophyll content of the leaves was calculated according to the following formula: [(0.0127 × A663 − 0.0629 × A645) + (0.0229 × A645 − 0.0468 × A663)] × total extract volume/fresh weight of the sample [17].

Proline content: Fresh leaves (0.1 g) were ground with 1.5 mL of 3% sulfosalicylic acid into a homogenate and then boiled for 15 min and centrifuged at 4000 rpm for 10 min. The 0.5 mL extract was mixed with 0.5 mL of glacial acetic acid and 0.5 mL of 25% ninhydrin. After boiling for 30 min, 1 mL of toluene was added to the mixture. The absorbance was measured at a wavelength of 520 nm. The proline content in the sample was calculated with the following formula: the proline content corresponding to the A520 in the standard curve × total volume of extract/total sample weight × volume of the measured extract [18].

### 2.3. RNA Extraction and Library Construction

Mature leaves were harvested at 0 d, 3 d, and 6 d of treatment. Total RNA was extracted using an RNAprep Pure Plant Kit (DP441) (Tiangen Biotech Co., Beijing, China). Reverse transcription was performed to obtain the first cDNA chain through FastKing RT Enzyme (KR116) (Tiangen Biotech Co., Beijing, China). The complementary strand was synthesized by adding switch oligo splices to the first cDNA strand. LongAmp-tagged PCR amplification was performed for 14 cycles. The PCR products were then connected to the ONT connector with T4 DNA ligase. The purification of the product was conducted with Agencourt XP magnetic beads. The cDNA library was constructed with the cDNA-PCR Sequencing Kit (SQK-PCS109) (Tiangen Biotech Co., Beijing, China) and sequenced using the PromethION platform (Biomarker Technologies Co., Beijing, China). Three biological replicates were assayed for each treatment.

### 2.4. RNA-Seq and Data Analysis

After sequencing, low-quality reads (less than 500 bp and Q-score less than 6) and ribosomal RNA reads were filtered from the raw data. The resulting full-length sequences were polished, and Minimap2-2.25 software [19] was used for comparison with the genome data of *S. matsudana* Koidz [20]; the redundant sequences were removed with Minimap2 software. The reads were visualized using Integrative Genomics Viewer (IGV), and the gene structure was optimized using Gffcompare v0.12.2 software. The resulting transcript was used for alternative splicing analysis. The AS type of each gene in the samples was obtained using the Astalavista v 4.0 software [21]. The AS sites and intron model were analyzed using the Stata 18 software by comparing them with the genome data. Gene expression levels were recorded via FPKM calculation using DESeq2 v 1.24.0 [22]. The differentially expressed genes (DEGs) were obtained considering fold change ≥ 2 and false discovery rate < 0.01. The gene sequences were compared with Gene Ontology (GO) to determine their functional annotations (http://geneontology.org/, accessed on 1 May 2022). Hypergeometric tests were used to identify pathways that were significantly enriched in target genes compared to the entire genetic background in the Kyoto Encyclopedia of

Genes and Genomes (KEGG) database (http://www.genome.ad.jp/kegg/, accessed on 1 May 2022).

### 2.5. RT–PCR Validation

The specific primers were designed at both sides of one or two AS events (Supplementary Material Table S1). RT–PCR was conducted to validate the AS event. The PCR system contained 10 μL of 2× EasyTaq PCR SuperMix, 7 μL of ddH$_2$O, 2 μL of a pair of primers, and 1 μL of the cDNA sample. The amplification reaction was subjected to initial denaturation at 94 °C for 5 min, followed by repeated denaturation at 95 °C for 1 min, annealing at 55 °C for 30 s, and elongation at 72 °C for 1 min, for a total of 35 cycles. The final elongation step was carried out at 72 °C for 10 min.

## 3. Results

### 3.1. Physiological Performance of S. matsudana Koidz 9901 under Salt Stress

Under salt treatment, *S. matsudana* Koidz 9901 leaves displayed gradual water loss, wilting, and yellowing. Serious damage was apparent after 6 d of salt stress, and the lower leaves started to fall off (Figure 1A). Meanwhile, the physiological data showed the corresponding changes. The chlorophyll content at 6 d decreased by 21.50% compared to that at 0 d, while the relative electrical leakage, MDA content, and proline content increased by 34.69%, 27.65%, and 30.35%, respectively. Comparably, most indexes did not change significantly (Figure 1B) at 3 d of salt treatment compared to 0 d, except for the proline content.

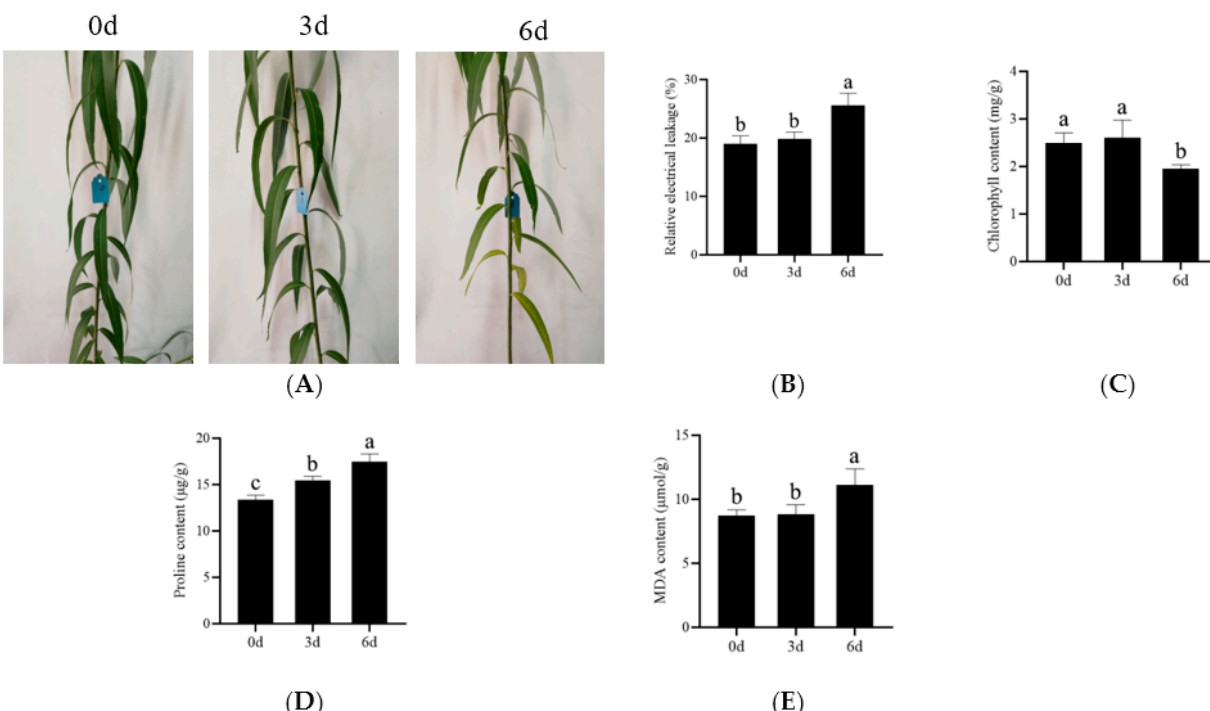

**Figure 1.** Phenotype (**A**) and physiological performance (**B–E**) of *S. matsudana* Koidz 9901 under salt treatments for 0 d, 3 d, and 6 d. The different letters in columns (**B–E**) indicate the significant differences at the level of $p < 0.05$ according to the LSD test (SPSS 24.0).

### 3.2. Overview of Iso-Seq Data

Full-length transcriptome sequencing was conducted for the salt treatment samples at 0 d, 3 d, and 6 d. A total of 56.7 GB of clean data were obtained, with an average of 6.30 GB per sample (Table 1). The full-length fragments accounted for 92.64% of the total and were annotated to the genome sequences of *S. matsudana* by 95.96%, indicating the high quality of the sequencing data.

**Table 1.** Sequence output of 9901 willow transcriptome and alignment with *S. matsudana* genome.

| Scheme * | Number of Clean Reads | Number of Full Length Reads | Full-Length Percentage | Number of Mapped Reads | Mapped Rates |
|---|---|---|---|---|---|
| 0-1 | 5,376,795 | 4,976,324 | 92.55% | 4,813,677 | 96.73% |
| 0-2 | 5,651,929 | 5,249,208 | 92.87% | 5,093,243 | 97.03% |
| 0-3 | 6,263,598 | 5,885,644 | 93.97% | 5,504,312 | 93.52% |
| 3-1 | 4,546,552 | 4,209,596 | 92.59% | 4,061,050 | 96.47% |
| 3-2 | 5,471,691 | 5,085,932 | 92.95% | 4,923,362 | 96.80% |
| 3-3 | 6,171,131 | 5,681,920 | 92.07% | 5,172,763 | 91.04% |
| 6-1 | 5,561,594 | 5,144,088 | 92.49% | 5,015,632 | 97.50% |
| 6-2 | 6,071,375 | 5,607,172 | 92.35% | 5,455,723 | 97.30% |
| 6-3 | 5,127,798 | 4,712,937 | 91.91% | 4,582,490 | 97.23% |

* Samples 0-1, 0-2, and 0-3 represent 3 replicates at 0 d salt treatment, and the other series are similarly designated.

All transcript sequences were aligned to the *S. matsudana* genome. After removing the redundant sequences via the Minimap2 software, 78,624 transcript sequences were obtained for 64,235 unique genes. A total of 14,389 transcripts were determined to have the AS isoforms, corresponding to 4786 AS genes. These genes accounted for 7.45% of the genes in the transcriptome library, and each AS gene produced an average of 3.01 isoforms.

### 3.3. AS Events in 9901 Willow under Salt Stress

AS events in 9901 willow under salt stress were classified into five types: alternative 3′ splice sites (A3SS), alternative 5′ splice sites (A5SS), exon skipping (ES), intron retention (IR), and mutually exclusive exon (MXE) (Figure 2A). Based on the statistical results, a total of 9307 AS events were determined in all sequencing samples, with the IR type as the most common, accounting for 44.15%–46.05% of all AS events, followed by A3SS, accounting for 24.84%–25.18%, and the MXE type, accounting for 0.54%–0.70% (Figure 2B). During salt treatment, the percentage of IR events decreased from 46.05% at 0 d to 44.15% at 6 d. However, the percentage of the other types increased. The MXE type increased gradually, A5SS and A3SS increased in earlier days (0 d–3 d) but changed less in later days (3 d–6 d), and ES remained stable in early days and increased in later periods of salt treatment.

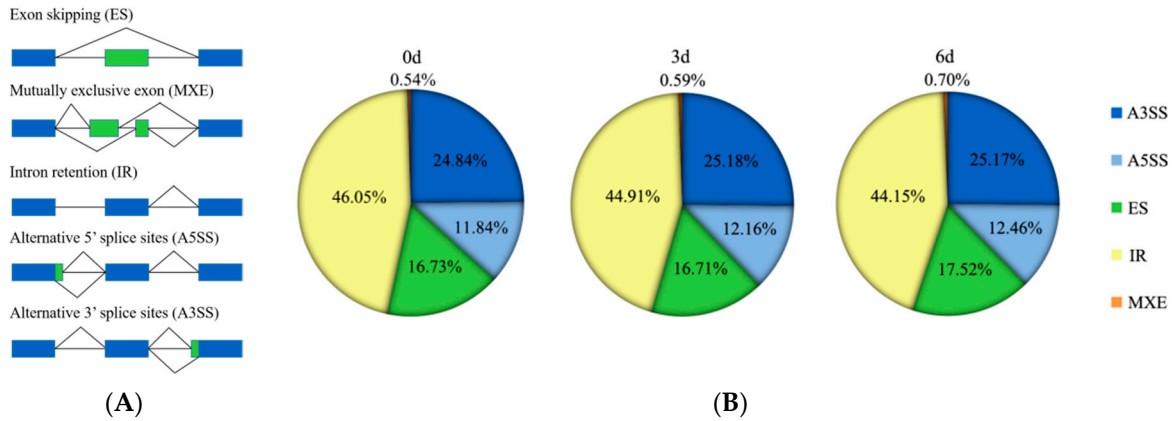

| (**A**) | (**B**) |
|---|---|

**Figure 2.** AS landscapes of 9901 willow based on Iso-Seq data: (**A**) AS types; the blue box indicates exons, the green box indicates exons that are alternatively spliced, the straight line indicates introns, and the broken lines indicate splice junctions; (**B**) proportion of the AS types at 0 d, 3 d and 6 d of salt treatment.

### 3.4. Alternative Splicing Modes of the 9901 Willow Genes

Based on the genome sequence data of *S. matsudana*, a total of 46,529 genes and their encoding sequences were predicted. The bases on both sides of introns were analyzed (site +1 and +2 at the 5′ site and −2 and −1 at the 3′ end), which showed that 5′GT-AG3′ was the

most common mode, with 205,691 introns and accounting for 89.09% of the total, followed by 5′GC-AG3′, with 9370 introns and accounting for 4.06% of the total (Figure 3A).

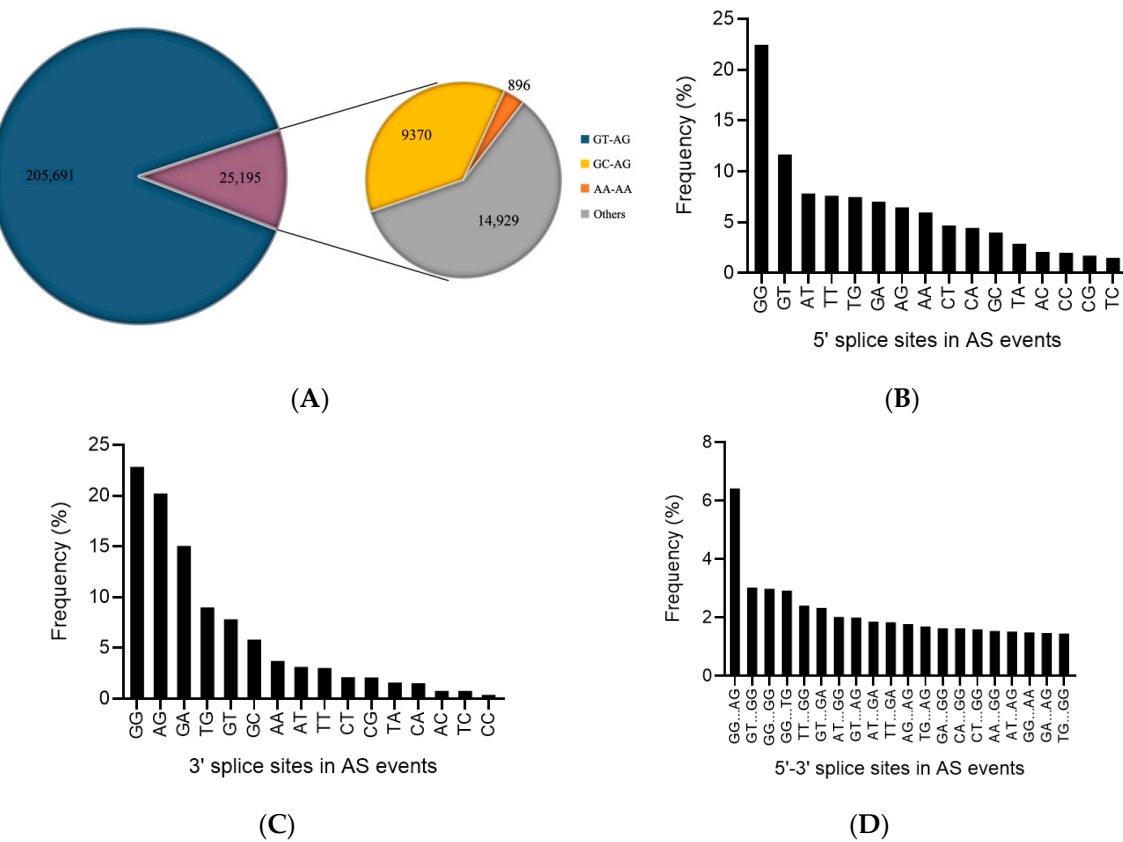

**Figure 3.** The intron splicing mode of the transcripts in willow: (**A**) the proportion of the intron splice mode of the predicted transcripts in the *S matsudana* genome; (**B–D**) the frequency of the splice motif at the 5′ site (**B**) and 3′ end (**C**) and mode (**D**) of the AS transcripts (except for IR events) in 9901 willow under salt treatment.

The 9307 AS events in the 9901 willow transcripts under salt stress consisted of 4991 unconventional splicing events (the others were IR events), which accounted for 230 splice patterns of introns. The most unconventional splice motif of introns at the 5′ site was 5′GG, with a frequency percentage of 22.46%, followed by 5′GT with a frequency percentage of 11.68% (Figure 3B). The most unconventional splice motif of introns at the 3′ end was GG3′, with a frequency percentage of 22.84%, followed by AG3′, with a frequency percentage of 20.22% (Figure 3C). Notably, 5′GG-AG3′ was the most common intron mode, with a frequency percentage of 6.43%, followed by 5′GT-GG3′, with a frequency percentage of 3.03% (Figure 3D).

### 3.5. Characterization of AS Genes

Based on the number of AS events that occurred in a gene, all genes were classified into three types: high-AS genes (AS events ≥ 5), low-AS genes (4 ≥ AS events ≥ 1), and non-AS genes. Distinguishably, 276 high-AS genes, 4510 low-AS genes, and 53,637 non-AS genes were identified in 9901 willow under salt treatment.

The characterization of three types of AS genes showed that the number of exons in high-AS genes was 1.32 times and 1.96 times greater than that in low-AS genes and non-AS genes, respectively. Meanwhile, the length of exons in high-AS genes was 1.32 times and 2.51 times longer than that in low-AS genes and non-AS genes; the length of introns in high-AS genes was 1.79 times and 3.43 times longer than that in low-AS genes and non-AS genes; and the expression abundance of high-AS genes was 1.46 times and 3.32 times higher

than that of low-AS genes and non-AS genes. Conversely, the GC content of non-AS genes was 1.04 and 1.03 times higher than that of high-AS genes and low-AS genes, respectively. These results suggest that the occurrence of AS events is significantly correlated with the gene sequence, structure, and expression level (Figure 4).

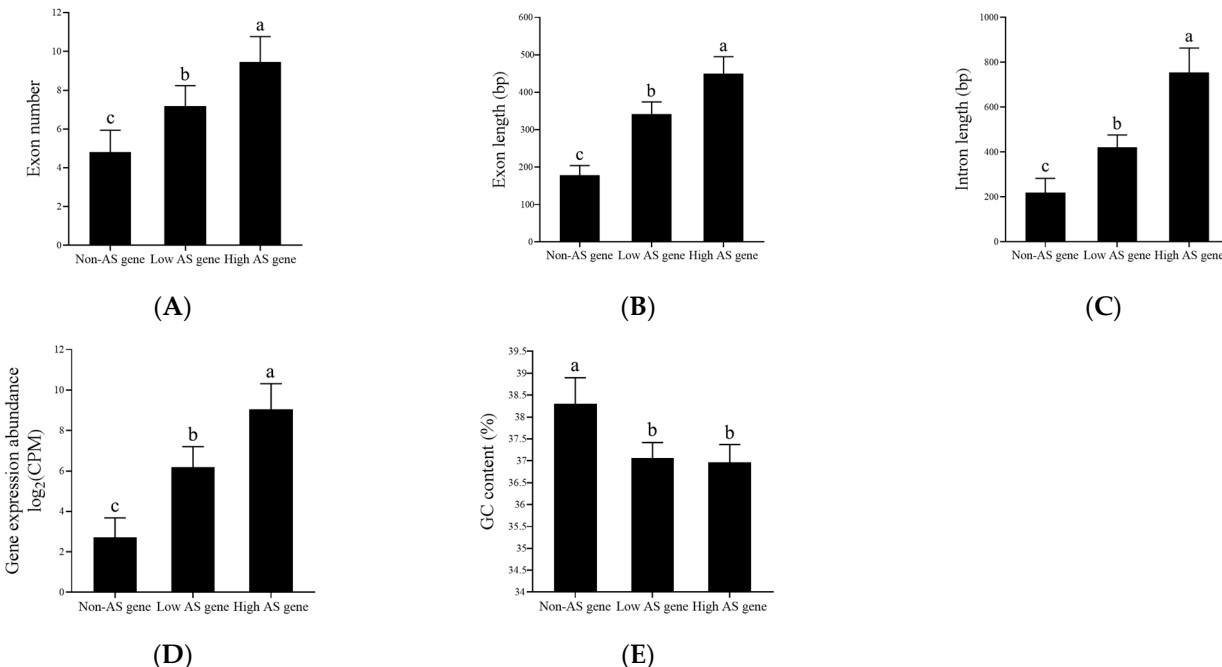

**Figure 4.** Multiple comparisons among high-AS genes, low-AS genes, and non-AS genes in the number of exons (**A**), length of exons (**B**), length of introns (**C**), gene expression abundance (**D**), and GC content (**E**). Error bar means standard deviation. The different letters indicate significant differences at the level of $p < 0.05$.

*3.6. AS Performance of Willow Genes during Salt Stress*

Statistical analysis revealed 2954, 2938, and 3399 AS genes in the samples after 0 d, 3 d, and 6 d salt treatment, respectively. Among the identified genes, 602 AS genes only occurred under normal growth conditions (0 d), 935 AS genes newly occurred in the earlier salt stress period (3 d) in comparison to that at 0 d, and 1333 AS genes newly occurred in the later salt stress period (6 d) in comparison to that at 0 d. A total of 436 AS genes were consistently present after salt stress (3 d and 6 d), and 1717 AS genes were consistently present in all samples (0 d, 3 d, and 6 d salt stress) (Figure 5A). These results indicate that an increasing number of genes underwent the AS process during salt stress, especially in the later period of salt treatment, which might exert more effects on plant performance.

GO annotations of the 935, 1333, and 436 AS genes that occurred only at 3 d, 6 d, and consistently at 3 d and 6 d salt stress were determined. All three groups had 59, 74, and 31 enriched genes in the RNA binding item, respectively (Figure 5B,D,F). KEGG analysis showed that the 935 gene group was significantly enriched in purine metabolism (16 genes) and RNA transport (22 genes) (Figure 5C), and the 1333 AS gene group was significantly enriched in autophagy (10 genes) (Figure 5E), whereas the 436 AS gene group was significantly enriched in porphyrin and chlorophyll metabolism (7 genes) (Figure 5G).

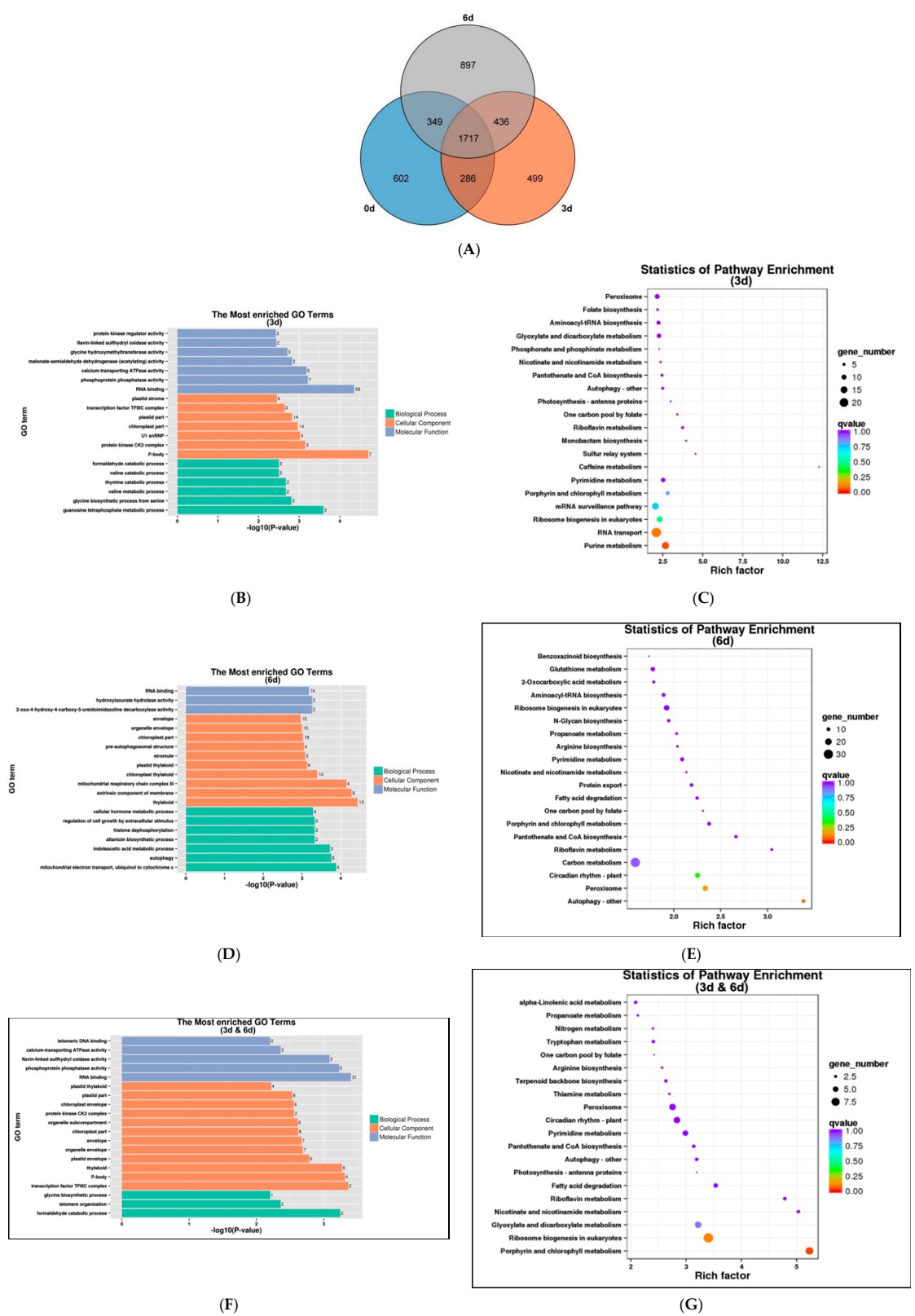

**Figure 5.** Annotation of AS genes during salt treatment: (**A**) Venn plot of AS genes in 9901 willow at 0 d, 3 d, and 6 d salt treatment; (**B**,**D**,**F**) GO enrichment of AS genes occurring only at 3 d, 6 d, and consistently at 3 d and 6 d; (**C**,**E**,**G**) KEGG enrichment pathways of AS genes occurring only at 3 d, 6 d, and consistently at 3 d and 6 d. GO analysis was conducted with the online database http://geneontology.org/ (accessed on 1 May 2022). KEGG analysis was conducted with the online database http://www.genome.ad.jp/kegg/ (accessed on 1 May 2022).

*3.7. AS Patterns of Five Genes during Salt Stress*

Based on the transcriptome analysis, five adversity stress-related genes were selected to determine their AS pattern during salt treatment. The RT–PCR results of AS events and expression abundance were consistent with the RNA-seq data. For example, two bands were obtained for the Arabidopsis pseudo-response regulator 2 gene (*APRR2*): One is a normal transcript showing an upregulation mode during salt stress, and the other is an isoform with the first intron retention (introducing a PTC) showing a downregulation pattern. The *ASK* gene encodes a member of the E3 ubiquitin ligase SCF complex. The primers were designed for the third intron retention event (introducing a PTC). Two bands were obtained at each period of salt treatment; the normal transcript was increased during salt stress, while the isoforms were decreased significantly. Additionally, one new AS event (the twelfth intron retention) was generated in the *ASK* gene at 3 d salt treatment, and two more AS events were found at 6 d salt treatment (A3SS of the first exon, the ninth intron retention). The first intron retention event test (introduce a PTC) of the photomorphogenesis transcription factor *HY5* revealed two bands; the normal transcript was downregulated during salt stress, while the isoform gradually increased. The protein phosphatase gene *PP2C* had the second and fifth intron retention events (both introduced a PTC) at 0 d of salt treatment. During salt treatment, the IR2 isoform was increased in expression, whereas the IR5 isoform was downregulated. Additionally, a new sixth intron retention event was generated in the *PP2C* gene due to salt stress. The stress osmotic/ABA-activated protein kinase gene *SAPK2* had sixth intron retention (introducing a PTC) and seventh exon A3SS events. The normal transcript and seventh exon A3SS isoform showed a downregulation during salt stress, while the sixth intron retention isoform displayed upregulation. Moreover, a new first intron retention event occurred in the *SAPK2* gene after salt stress (Figure 6). These assays indicate that the genes underwent dynamic changes in AS events with salt treatment, not only in terms of their expression levels but also in terms of an increase in the number of AS sites and types, which resulted in diverse transcripts and synergistically regulated the response of willow to salt stress.

*3.8. Evaluation of Differentially Expressed Genes (DEGs) and Differential AS (DAS) Genes against Salt Stress in 9901 Willow*

Comparing the transcript data of 3 d and 6 d of salt stress with the data of 0 d, 1160 and 2332 differentially expressed genes (DEGs) were found, respectively. Clearly, a longer exposure to salt stress resulted in more gene expression differences. With both salt stress treatments combined, a total of 2764 DEGs were identified, 1346 of which were upregulated and 1418 were downregulated. Meanwhile, 1886 and 2221 differentially expressed AS (DAS) genes were identified at 3 d and 6 d of salt stress compared to 0 d, respectively. Similarly, a longer exposure to salt stress resulted in more AS genes. With both salt stress treatments combined, a total of 3069 genes underwent differential AS due to salt stress.

Comparing DEGs with DAS genes, fewer genes were shared. For instance, 142 shared genes were found at 0 d vs. 3 d, accounting for 12.24% of the 1160 DEGs and 7.53% of the 1886 DAS genes. Meanwhile, 264 shared genes were found at 0 d vs. 6 d, accounting for 11.32% of the 2332 DEGs and 11.89% of the 2221 DAS genes. Fifty-two shared genes were found at 3 d vs. 6 d, accounting for 8.58% of 606 DEGs and 2.56% of 2031 DAS genes (Figure 7A).

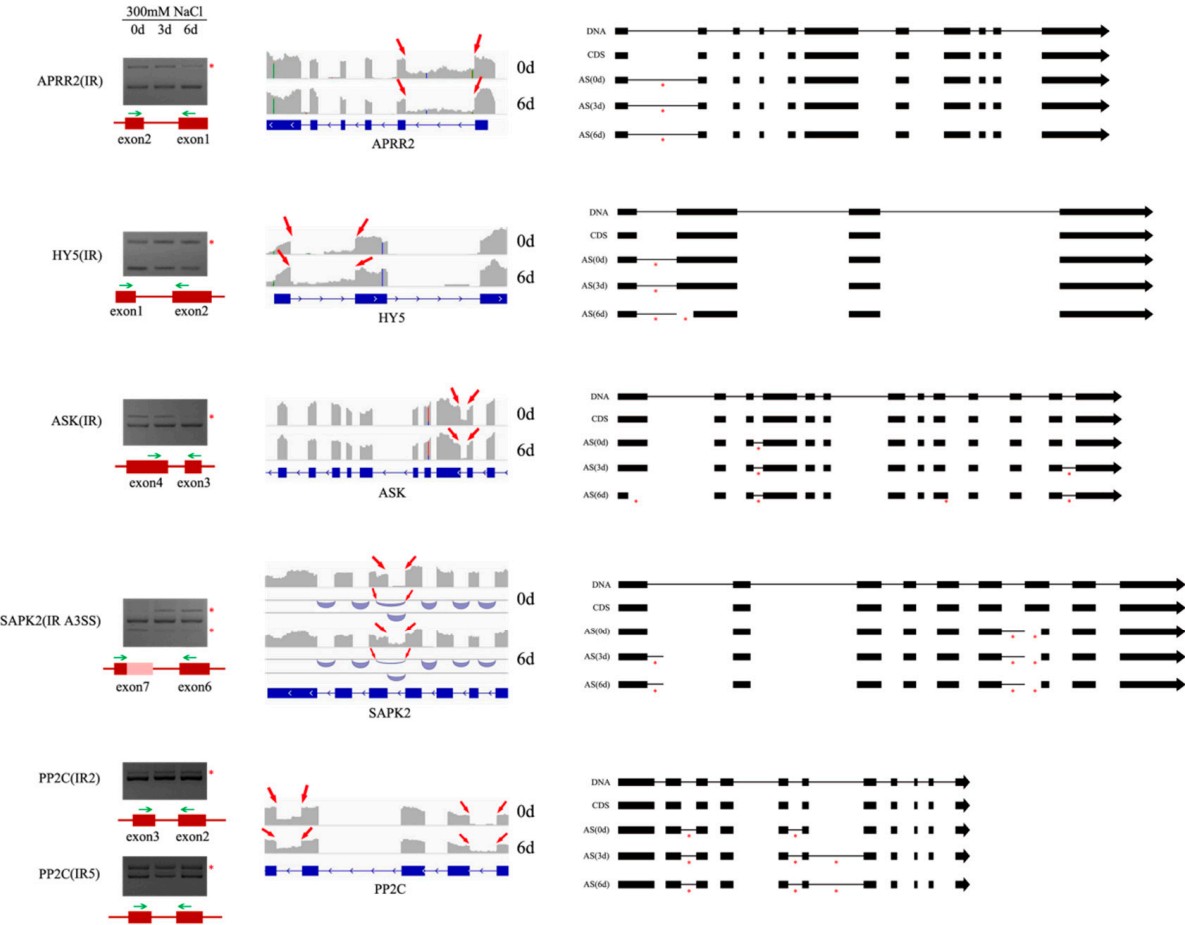

**Figure 6.** AS events of five genes during salt stress. The left section shows RT–PCR results; the red star denotes the alternative splice form. The schematic figure is below each gel image, with the block indicating an exon, the line representing an intron, and the pink block representing an A3SS event. The green arrows indicate primer locations for AS events. The middle section presents the visualization of the transcriptome results via the IGV browser; the grey peaks indicate the RNA-seq read density across the gene. The red arrows represent alternative splice sites. The blue arcs indicate splice junctions. The relative gene structure is given below each panel. The right section illustrates the diagram of AS events at 0 d, 3 d, and 6 d of salt treatment based on RNA-seq data; the red star represents the AS events that occurred.

KEGG analysis showed that the 1160 DEGs at 0 d vs. 3 d were significantly enriched in the pathways of glyoxylate and dicarboxylate metabolism, carbon metabolism, and carotenoid biosynthesis, while the 1886 DAS genes were significantly enriched in the pathways of the purine metabolism pathway. In addition, the 2332 DEGs at 0 d vs. 6 d were significantly enriched in the pathway of flavonoid biosynthesis, carbon metabolism, and carbon fixation, while the 2221 DAS genes were significantly enriched in the pathways of autophagy, carbon metabolism, pantothenate, and CoA biosynthesis. A total of 606 DEGs at 3 d vs. 6 d were significantly enriched in the pathways of vitamin B6 metabolism and photosynthesis, while the 2031 DAS genes were not significantly enriched in any pathways (Figure 7B,C). Obviously, both DEGs and DAS genes significantly contribute to the regulation of willow resistance to salt pressure, but through different pathways. Mostly, a gene was preferentially regulated only through one or the other pathway during salt stress.

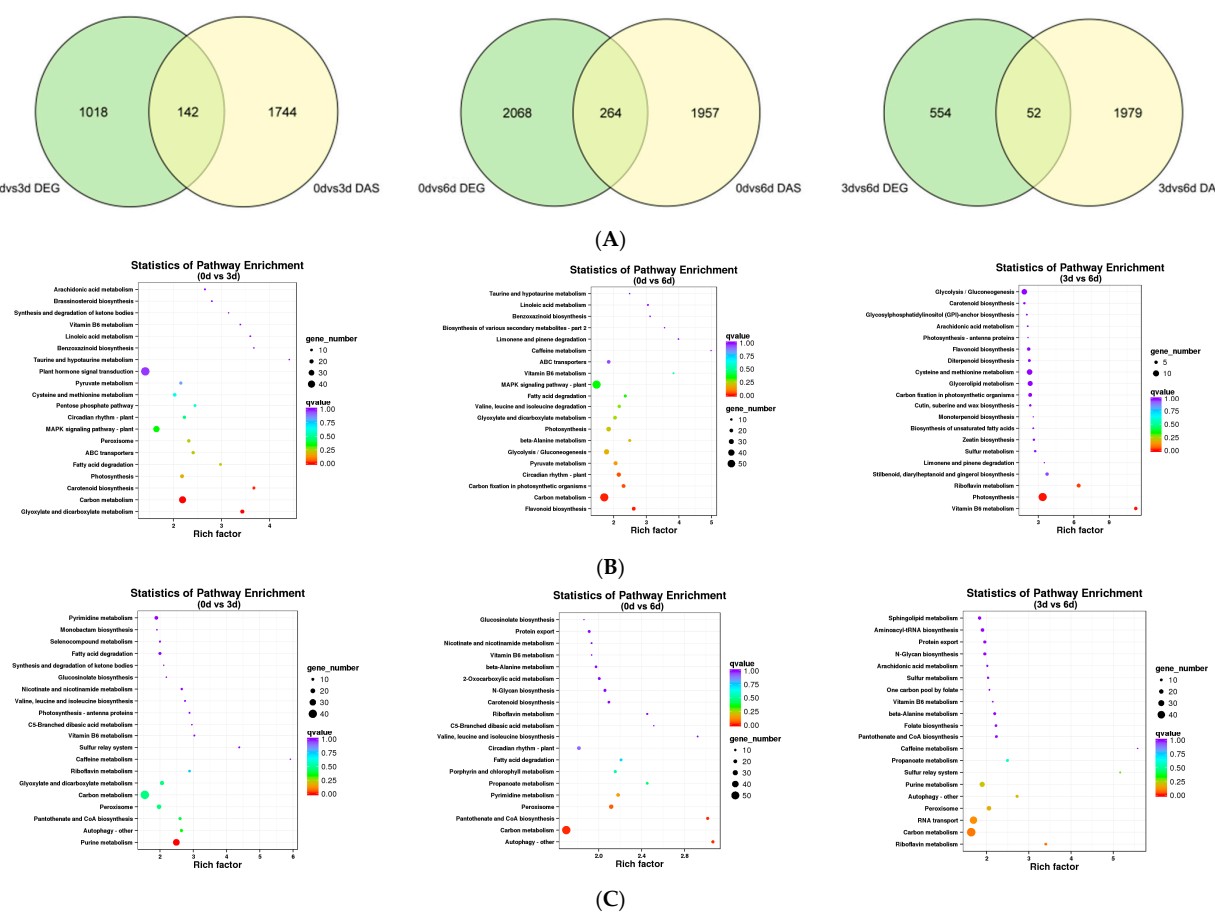

**Figure 7.** Comparison between DAS genes and DEGs during salt stress: (**A**) Venn plots of DEGs and DAS genes at 0 d vs. 3 d, 0 d vs. 6 d, and 3 d vs. 6 d; (**B**) KEGG enrichment pathway of DEGs at 0 d vs. 3 d, 0 d vs. 6 d, and 3 d vs. 6 d; (**C**) KEGG enrichment pathway of DAS genes at 0 d vs. 3 d, 0 d vs. 6 d, and 3 d vs. 6 d.

## 4. Discussion

### 4.1. Characterization of AS Genes in Willow

AS events are widespread in plant transcripts. For example, 14,172 AS events were identified in 6798 genes (31.58% of the multiexon genes) in *Gossypium davidsonii* [23]. A total of 3810 AS genes (9225 AS events) were identified in *P. alba* × *P. glandularulosa* cv.84K poplar, accounting for 13.51% of all expressed genes [24]. Our study involved the AS analysis of willow. A total of 4786 AS genes were identified, accounting for 7.45% of all expressed genes. It was concluded that AS is an important mode of gene expression regulation.

Among the AS events of willow genes, the IR type accounted for the most (46.05%) of all AS events. Several species showed similar results; for instance, IR types accounted for 38.7% of all AS events in *P. alba* var. *Pyramidalis* [25] and 53% in *cassava* [8], indicating the prevalence of IR events. It is possible that introns play an important role in AS events and diversify gene function. However, the percentage of each AS type differed in plant species [5]. For example, the percentage of ES events (16.73%) was greater than that of A5SS (11.84%) in willow, which is consistent with the findings in rice [26,27]. However, in poplar, they were 8.68% and 23.71%, respectively [24].

Additionally, we found that 5′GG and AG3′ were the most common motifs for the AS sites, with 22.46% and 22.84% in willow, respectively, and 5′GG-AG3′ was the most common mode with 6.43%. In poplar, 5′AA and CA3′ were highly recognized as AS sites [28]. The differences in AS motifs in species might explain the reasons for their AS-type variation. Even in *Arabidopsis thaliana*, adjacent bases might affect splice sites. For example, AAG is more prevalently attached to the 5′GT and 5′GC splice motifs, but

5′GTAAG accounts for only 17.17% of the 5′GT motif, and 5′GCAAG accounts for 58.06% of the 5′GC motif. Notably, 5′GCAAG prefers AG3′ as its intron mode, while 5′GTAAG does not have a strong motif bias at the 3′ end [29]. Thus, the flanking bases of the motif also influence the occurrence of AS events. Furthermore, some gene characteristics such as exon number, exon and intron lengths, transcript levels, and GC content were found to be significantly associated with AS events in poplar, maize, and soybean [5,24,30]. Our results also support this point. This suggests that differences in gene composition, gene structure, and expression have an effect on splice site motifs and lead to the diversity of AS genes in different species. In a previous study, a total of 71 conserved AS events were identified in *Populus* and *Eucalyptus* (1.2% and 2.4% of the total AS events, respectively) [7]. Thus, the AS pattern can be an important indicator for species.

### 4.2. Abiotic Stress Advanced the AS Events as a Mechanism of Plant Decline

It has been shown that stress affects splice-related genes and their regulators through AS, leading to the recognition of splice sites in bias [31–33]. AS led to a decrease in the normal transcript abundance, further affected the physiological activities in cells, and reduced plant growth vigor under stress. In other reports, salt stress induced 2065 AS events in 1088 genes in *Arabidopsis* [34] and induced 1466 AS events in grape roots [35]. In wheat seedlings, AS events were induced in 200, 3576, and 4056 genes under drought, heat, and combined stress, respectively [36]. In our study, 935 genes had new AS events in willow at 3 d of salt stress, and 897 more genes had new AS events at 6 d of salt stress. Obviously, adversity stress greatly promotes AS occurrence.

Adversity stress also led to changes in AS types, especially exons. A report on *Brassica oleracea* L. showed that both A5SS and A3SS types increased from 12% to 13%, and ES types increased from 7% to 16% under high temperatures [37]. In *P. tomentosa* Carr., the A5SS, A3SS, and ES types increased from 26.14% (25 °C) to 26.45% (40 °C), from 19.6% to 21.02%, and from 9.54% to 10.22%, respectively [15]. In this study, the A5SS, A3SS, and ES types in willow increased from 11.84% (0 d) to 12.46% (6 d), from 24.84% to 25.17%, and from 16.73% to 17.52%, respectively. Changes in exons significantly lead to the heterogeneity of the encoded proteins in structure and further reduce or lose their function. For example, the drop-off of a part of the exon in the isoform *ZmPP2C26S* of maize resulted in the loss of a motif interacting with MAPK, which led to a loss of the dephosphorylation of ZmMAPK7 [38]. In our study of willow, *HY5*, *ASK*, and *SAPK2* all underwent exon excision events. With a longer duration of salt treatment, more AS events and AS types were observed. Further exploration of their effects on function is necessary.

### 4.3. AS Effects on Salt Resistance of Plants

Gene function annotation revealed that AS genes in 9901 willow under salt stress were primarily related to RNA binding. Several studies have confirmed that RNA binding proteins (RBPs) are extensively involved in post-transcriptional regulation, such as RNA splicing, translocation, sequence editing, intracellular localization, and translational control, and influence plant resistance to stress [39]. For example, *SbGR-RNP* in *Sorghum bicolor* was increased by 4–7-fold under NaCl treatment [40]. The overexpression of *LbGRP1* from *Limonium bicolor* increased superoxide dismutase and catalase activities in transgenic tobacco under salt stress and significantly improved plant salt tolerance [41]. In our study, *SmGRP1* did not have AS events at 0 d and 3 d of salt treatment but underwent AS at 6 d, which accordingly led to a decrease in the related enzymes and resulted in serious damage. Another gene, *SmRNP1*, exhibited a reverse trend. It underwent AS at 0 d and 3 d of salt treatment but recovered at 6 d. A previous study showed that the overexpression of *AtRNP1* could seriously downregulate the expression level of some stress-responsive genes such as *RD29A*, *RD29B*, *KIN1*, and *DREB2A*, and then reduce plant tolerance to drought, salt, osmosis, and ABA stress [42]. Therefore, the AS performance of *SmRNP1* exhibited a similar pattern to the pattern of salt resistance in willow plants, which is in agreement with the physiological data of willow under salt stress.

KEGG analysis revealed the occurrence of 935 AS genes of willow at 3 d of salt stress, which were significantly enriched in the purine metabolic pathway (16 AS genes). A study reported that the *Tibetan hulless* barley cultivar 0119 tends to reduce nucleotide metabolism to conserve energy in the early stage of salt stress, which can maintain plant growth [43]. In *Arabidopsis*, a mutation in adenine phosphoribosyl transferase (APT1) caused the conversion of adenine to adenosine monophosphate (AMP) to fail. The mutant plants then accumulated more adenine content and displayed better resistance to high-temperature stress [44]. Therefore, the occurrence of AS in purine-metabolism-related genes is probably advantageous for plants, which also corresponds to the better performance of willow in the earlier days of salt stress.

In addition, 436 willow genes consistently had AS events during salt stress (3 d and 6 d) and were mostly enriched in the porphyrin and chlorophyll metabolic pathways (7 AS genes). Among these genes, *SGR2* is recognized as an important gene involved in the greening of plant leaves and fruits. *Arabidopsis* plants overexpressing *SGR2* exhibited a stay-green phenotype due to the synthesis of more chlorophyll [45]. The *SmSGR2* gene in willow had one AS event at 3 d of salt stress and three AS events at 6 d of stress. Protoporphyrinogen oxidase (PPOX1) is a key enzyme in the chlorophyll biosynthesis pathway. The overexpression of the *MhPPOX1* gene in tobacco and *Arabidopsis* plants displayed significantly higher chlorophyll content than wild-type plants. In our data, *SmPPOX1* had one AS event at 3 d of salt stress and two AS events at 6 d of stress [46]. Obviously, with a longer duration of salt stress, more AS events occurred in the *SmSGR2* and *SmPPOX1* genes, which accordingly resulted in a decrease in the normal transcripts and further affected the synthesis of chlorophyll. This result is in agreement with the lowest chlorophyll content observed at 6 d of salt stress. Therefore, the AS process of the genes involved in porphyrin and chlorophyll metabolism deserves further investigation.

A total of 1333 AS genes were observed in willow after 6 d of salt treatment and were significantly enriched in the autophagy pathway (10 AS genes). Cellular autophagy is an important protein degradation pathway in plants that helps plants remove damaged proteins and organelles from the cell and therefore plays an active role in plant resistance to adversity stress [47]. Thus, the AS process in autophagy-related genes might cause inefficient degradation of excess cellular metabolites, which then affects the regular cellular activity and results in serious damage in willow. For example, the PP2C-type phosphatase gene *Ptc2* had AS events at 6 d only. A previous study showed that this gene plays a key role in initiating autophagy by dephosphorylating the Ser429 site of *ATG13* and is important for plant growth [48]. Further exploration of the function of AS isoforms would be significant to understand the possible effects.

In summary, thousands of willow genes underwent AS events under salt stress, which greatly increased the diversity of mRNAs expressed from the genome. They may regulate the expression level of normal transcripts and influence their contribution to the stress resistance of plants. Additionally, the possible protein isoforms may have different localization, varied enzymatic properties, and different interactions with proteins and nucleic acids; they may also have functional effects on morphological and physiological performance. Therefore, characterizing the function of isoforms is necessary to understand salt resistance mechanisms in willow. Controlling the AS events of genes under stress is of great significance in improving plant resistance. Additionally, our results show that AS is an independent and distinct way of regulating plant resistance from differentially expressed genes. We conclude that AS and DEGs collaborate in the regulation of salt resistance in willow.

## 5. Conclusions

A full-length transcriptome analysis revealed that AS events were widely present in willow trees. With a longer duration of salt stress, the number and types of AS events markedly increased. AS genes were mainly enriched in the RNA binding and participated in purine metabolism, porphyrin and chlorophyll metabolism, and the autophagy pathway.

The results suggest that AS is widely involved in plant resistance to salt stress through a different mechanism from differential gene expression. Additionally, AS events in willow mostly occurred in the genes with longer and more introns and exons, low GC content, and high expression levels. Further exploration of the function of isoforms would be beneficial for better understanding the resistance mechanism of willow plants.

**Supplementary Materials:** The following supporting information can be downloaded at: https://www.mdpi.com/article/10.3390/f15010030/s1, Table S1: The specific primer sequences designed for AS events.

**Author Contributions:** X.W. and L.G.: investigation, writing—original draft preparation; J.Z. and L.W.: methodology, conceptualization; D.W.: writing—review and editing; J.X.: supervision. All authors have read and agreed to the published version of the manuscript.

**Funding:** This work was supported by the National Natural Science Foundation of China (grant number No. 31870648).

**Data Availability Statement:** Data will be made available upon request.

**Conflicts of Interest:** The authors declare no conflict of interest.

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
