# Peer review of "The Genome-Wide Profiling of Alternative Splicing in Willow under Salt Stress"

_forests, doi:10.3390/f15010030_

Round 1
Reviewer 1 Report
Comments and Suggestions for Authors
The manuscript titled 'Genome-wide profiling of alternative splicing in willow under salt stress' offers a promising contribution to the field by investigating the interplay between salt stress and alternative splicing in willow. The introduction and discussion are well written. The authors' comprehensive approach whilst integrating appropriate genomic analyses provides a valuable resource in plant stress biology.
The following minor corrections should be taken into consideration:
Line 54, 354 and 434 - replace 'adversity' with 'stress'
Line 84 - What was the total volume of the soil media per pot?
Line 85 - What was the actual macro- and micro-nutrient content of the 'nutrient soil'?
Line 87 - 300 mM NaCl seems rather high; how did the authors decide on this concentration, and what was the final NaCl concentration in the pots after adding the NACl? Were the pots at field capacity already at the point when the NaCl was added? We need more detail on this section.
Line 439 - please replace the word 'item'
Author Response
Line 54, 354 and 434 - replace 'adversity' with 'stress'
Reply: all were corrected
Line 84 - What was the total volume of the soil media per pot?
Reply: The pot is 30 cm in height and 20 cm in dimeter and full of soil media. Description was modified.
Line 85 - What was the actual macro- and micro-nutrient content of the 'nutrient soil'?
Reply: It is a mixture without nutrient content in detail. We attached the company name.
Line 87 - 300 mM NaCl seems rather high; how did the authors decide on this concentration, and what was the final NaCl concentration in the pots after adding the NACl? Were the pots at field capacity already at the point when the NaCl was added? We need more detail on this section.
Reply: As we described in introduction section, Salix matsudana Koidz 9901 is a highly salt-tolerant willow variety that is widely planted in saline-alkali land. We did the test and found 9901 willow had gradual changes under 300 mM NaCl treatment in a week. NaCl solution was added once until the sample harvest at 6th day (500 ml per pot and discard the leaking solution, which fully saturated the medium). Description was modified.
Line 439 - please replace the word 'item'
Reply: It was corrected
Reviewer 2 Report
Comments and Suggestions for Authors
Dear Editor in chief of Forest Journal
Hi and have a good day.
I reviewed the article entitled “Genome-Wide Profiling of Alternative Splicing in Willow 2 under Salt Stress” written by Jichen et al. This MS is written in good shape and have some novelty; however, it needs minor corrections for improve the article quality.
Introduction part:
Please refer the types of AS model in plants by diagram and explain in one paragraph.
There are many types of AS in plants but only the writers refer to intron retain. In this part all of AS must be reviewed by the writers.
Material and methods part:
In salinity stress with NaCl, it is better to use CaCl2 with NaCl to prevent CaCl2 deficit during salt stress. In line 84-87, I did not see this object.
Line 93-97: Please give references about measuring of Relative electrical leakage (REL) at the end of paragraph. This method is general method and I think another methods will be better than it.
Line 98-103: please give references about measuring of MDA at the end of paragraph.
Line 104-108: please give references about measuring of Chlorophyll content at the end of paragraph.
Please give references for proline measuring.
Line 148: please refer to final extension in PCR.
Result part:
Line 161: we do not have multiple comparisons. Please say which method used for mean comparisons like Tukey, LSD, and Duncan and so on.
Figure 4: the class of data is not correspond with error bar. Please check and correct them.
Figure 5: please refer which sites or software used to obtain GO and KEEG pathway in front of each part in the figure.
Discussion part:
In the result, the writers of the article obtain general information on AS and other thing for example 2764 DEGs and 1886 DASs ,….. How do you explain molecular mechanism of salinity in willow tree with many genes involved in salt stress? Please explain in one paragraph.
Author Response
Introduction part:
Please refer the types of AS model in plants by diagram and explain in one paragraph.
Reply: It was added.
There are many types of AS in plants but only the writers refer to intron retain. In this part all of AS must be reviewed by the writers.
Reply: The others were added.
Material and methods part:
In salinity stress with NaCl, it is better to use CaCl2 with NaCl to prevent CaCl2 deficit during salt stress. In line 84-87, I did not see this object.
Reply: In our experiment, we used nutrient soil as growing medium. We had test it with lower concentration of NaCI treatment for several willow varieties. The resistant variety like 9901 willow grew well and the sensitive variety displayed relatively bad. We refer that the nutrient soil is enough to support willow growing well under salt stress without adding the others.
Line 93-97: Please give references about measuring of Relative electrical leakage (REL) at the end of paragraph. This method is general method and I think another methods will be better than it.
Reply: The reference was added
Line 98-103: please give references about measuring of MDA at the end of paragraph.
Reply: The reference was added
Line 104-108: please give references about measuring of Chlorophyll content at the end of paragraph.
Reply: The reference was added
Please give references for proline measuring.
Reply: The reference was added
Line 148: please refer to final extension in PCR.
Reply: More details were added.
Result part:
Line 161: we do not have multiple comparisons. Please say which method used for mean comparisons like Tukey, LSD, and Duncan and so on.
Reply: More details were added.
Figure 4: the class of data is not correspond with error bar. Please check and correct them.
Reply: We did check and give more description.
Figure 5: please refer which sites or software used to obtain GO and KEEG pathway in front of each part in the figure.
Reply: More details were added.
Discussion part:
In the result, the writers of the article obtain general information on AS and other thing for example 2764 DEGs and 1886 DASs ,….. How do you explain molecular mechanism of salinity in willow tree with many genes involved in salt stress? Please explain in one paragraph.
Reply: More descriptions were added